# Effects of Acute and Chronic Resistance Exercise on the Skeletal Muscle Metabolome

**DOI:** 10.3390/metabo12050445

**Published:** 2022-05-16

**Authors:** Sebastian Gehlert, Patrick Weinisch, Werner Römisch-Margl, Richard T. Jaspers, Anna Artati, Jerzy Adamski, Kenneth A. Dyar, Thorben Aussieker, Daniel Jacko, Wilhelm Bloch, Henning Wackerhage, Gabi Kastenmüller

**Affiliations:** 1Department for Biosciences of Sports, Institute of Sport Science, University of Hildesheim, 31139 Hildesheim, Germany; 2Institute of Cardiovascular Research and Sports Medicine, German Sport University, 50933 Cologne, Germany; ausska@gmx.de (T.A.); d.jacko@dshs-koeln.de (D.J.); w.bloch@dshs-koeln.de (W.B.); 3Institute of Computational Biology, Helmholtz Zentrum München, German Research Center for Environmental Health, 85764 Neuherberg, Germany; patrick.dreher@helmholtz-muenchen.de (P.W.); werner.roemisch@helmholtz-muenchen.de (W.R.-M.); 4Department of Human Movement Sciences, Faculty of Behavioural and Movement Sciences, Amsterdam Movement Sciences, Vrije Universiteit Amsterdam, 1081 BT Amsterdam, The Netherlands; r.t.jaspers@vu.nl; 5Metabolomics and Proteomics Core, Helmholtz Zentrum München, German Research Center for Environmental Health, 85764 Neuherberg, Germany; anna.artati@helmholtz-muenchen.de; 6Institute of Experimental Genetics, Helmholtz Zentrum München, German Research Center for Environmental Health, 85764 Neuherberg, Germany; adamski@helmholtz-muenchen.de; 7Department of Biochemistry, Yong Loo Lin School of Medicine, National University of Singapore, Singapore 117597, Singapore; 8Institute of Biochemistry, Faculty of Medicine, University of Ljubljana, 1000 Ljubljana, Slovenia; 9Metabolic Physiology, Institute of Diabetes and Cancer, Helmholtz Zentrum München, German Research Center for Environmental Health, 85764 Neuherberg, Germany; kenneth.dyar@gmail.com; 10German Center for Diabetes Research (DZD), 85764 Neuherberg, Germany; 11Department of Sport and Health Sciences, Technical University of Munich, 80809 Munich, Germany; henning.wackerhage@tum.de

**Keywords:** resistance exercise, metabolomics, skeletal muscle, hypertrophy, beta-citrylglutamate, skeletal muscle adaptation, chenodeoxycholate

## Abstract

Resistance training promotes metabolic health and stimulates muscle hypertrophy, but the precise routes by which resistance exercise (RE) conveys these health benefits are largely unknown. Aim: To investigate how acute RE affects human skeletal muscle metabolism. Methods: We collected vastus lateralis biopsies from six healthy male untrained volunteers at rest, before the first of 13 RE training sessions, and 45 min after the first and last bouts of RE. Biopsies were analysed using untargeted mass spectrometry-based metabolomics. Results: We measured 617 metabolites covering a broad range of metabolic pathways. In the untrained state RE altered 33 metabolites, including increased 3-methylhistidine and N-lactoylvaline, suggesting increased protein breakdown, as well as metabolites linked to ATP (xanthosine) and NAD (N1-methyl-2-pyridone-5-carboxamide) metabolism; the bile acid chenodeoxycholate also increased in response to RE in muscle opposing previous findings in blood. Resistance training led to muscle hypertrophy, with slow type I and fast/intermediate type II muscle fibre diameter increasing by 10.7% and 10.4%, respectively. Comparison of post-exercise metabolite levels between trained and untrained state revealed alterations of 46 metabolites, including decreased N-acetylated ketogenic amino acids and increased beta-citrylglutamate which might support growth. Only five of the metabolites that changed after acute exercise in the untrained state were altered after chronic training, indicating that training induces multiple metabolic changes not directly related to the acute exercise response. Conclusion: The human skeletal muscle metabolome is sensitive towards acute RE in the trained and untrained states and reflects a broad range of adaptive processes in response to repeated stimulation.

## 1. Introduction

Resistance exercise (RE) with moderate to high loads stimulates muscle hypertrophy, increases strength and improves metabolic health [1]. Resistance training-associated stimuli activate the mTORC1 complex [2,3] and other signalling pathways [4] which orchestrate the synthesis of myofibrillar, sarcoplasmic and mitochondrial proteins for up to 72 h post-exercise [5]. Muscle dry mass is composed of ≈70% protein [6], and an elevation of protein synthesis above breakdown is essential to increase muscle mass [7]. However, proteins are not the only macromolecules synthesized in a hypertrophying muscle. The synthesis of the remaining 30% of muscle biomass relies on biosynthetic pathways that catalyse the synthesis of nucleotides required for RNA, DNA as well as phospholipids for cell membranes and signalling substrates [8]. Moreover, the rate of ATP hydrolysis increases rapidly at the onset of acute RE [9] and nucleotide turnover is increased during RE [10]. While the mechanisms regulating skeletal muscle protein synthesis in response to acute RE are well characterised, far less is known about how acute and chronic RE impacts the skeletal muscle metabolome, which includes a wide range of biomolecules involved in anabolism, catabolism and energy metabolism.

Huffman et al., determined gene expression and metabolite signatures in skeletal muscle biopsies after six months of endurance or combined endurance and strength exercise in older and younger subjects [11]. In their targeted metabolomics approach 15 amino acids, 45 acylcarnitines and seven organic acids were determined. Endurance exercise (EE) regimens with the highest energy turnover and also combined EE/RE produced the greatest increase in muscle acylcarnitines after the 6-month training period. This increase correlated with the expression levels of genes associated with mitochondrial energy metabolism. Moreover, the authors found a significant correlation between the training-induced change in acylcarnitine levels and the increase in oxygen uptake, indicating that acylcarnitine levels in muscle are linked to global aerobic capacity in humans.

Fazelzadeh and colleagues analysed the skeletal muscle and blood plasma metabolomes of 36 old and 24 young subjects at baseline and six months after whole body RE [12]. In muscle biopsies taken at rest, they determined the concentrations of 96 metabolites using different targeted metabolomics platforms. Metabolites covered by their approach included amino acids (and other amines), acylcarnitines, organic acids, oxylipins and nucleotides. A key finding was an increase in metabolites associated with the branched chain amino acids (BCAA) in younger subjects and a reduced catabolic BCAA-metabolism in older subjects. In general, mitochondrial metabolites were reduced in the older subjects. Interestingly, the authors found only a moderate correlation between the concentrations of metabolites measured in plasma and muscle, emphasizing the importance of analysing muscle biopsies for gaining insight into exercise-induced metabolic changes of skeletal muscle metabolites.

Results from these studies have revealed first insights into intramuscular alterations of the skeletal muscle metabolome after long-term exercise. However, the range of metabolites measured is limited and to date no study has measured acute changes of the skeletal muscle metabolome in untrained versus trained subjects. 

For the present study our aim was to uncover global alterations of the human skeletal muscle metabolome in response to acute RE in untrained and trained conditions using a non-biased, untargeted metabolomics approach. We therefore investigated how metabolites and related metabolic pathways are altered by acute and long-term, chronic RE training regimens known to induce mild hypertrophy in skeletal muscle.

## 2. Results

Skeletal muscle biopsies were collected from vastus lateralis muscles of six male participants to investigate acute changes of the muscle metabolome before and after an intense bout of resistance exercise (RE) under untrained and trained conditions. Biopsies were taken at rest, 45 min after a first bout of RE in an untrained state, and in a trained state, 45 min after the last bout of a five-week-long resistance training program (Figure 1).

Type I and type II fibre diameters increased significantly by 10.7 ± 4.1% and 10.4 ± 4.0%, respectively (Figure 2), suggesting that the exercise intervention had induced muscle hypertrophy.

We then performed an untargeted LC-MS analysis of all muscle biopsies. In total, we analysed the levels of 617 metabolites that were detected in at least 30% of the muscle biopsies. Of these 617 metabolites, 488 could be annotated with a known chemical structure. The identified metabolites cover a broad range of different metabolite classes including 136 amino acids and derivatives, 45 peptides, 36 nucleotides and derivatives, 17 carbohydrates, 5 energy metabolites, 176 lipids, 52 xenobiotics, 21 cofactors and vitamins (Figure 3A). In addition, we measured the abundance of 129 so-called unknown metabolites, representing molecules with yet unidentified chemical structures (denoted by the letter X followed by a number as the compound identifier; see Appendix A). First, we tested all 617 metabolites for their changes in response to acute exercise by comparing the concentrations of metabolites in the skeletal muscle before and after one bout of RE under untrained conditions. In a second step, we tested the metabolites for their changes in response to chronic training by comparing metabolite levels after a bout of exercise before and after 5 weeks of resistance training.

### 2.1. Effects of Resistance Exercise on the Acute Response of the Skeletal Muscle Metabolome

We aimed to identify metabolites that changed their concentration in skeletal muscle in response to acute unaccustomed RE. By using paired *t*-tests, we compared their intramuscular levels at 45 min after the first bout of resistance exercise with those at rest. We found 17 metabolites that increased after acute RE and 16 metabolites whose levels decreased (*p* < 0.05) (5 increased after acute and chronic training (Figure 3B intersection)). These 33 metabolites are distributed over 6 out of the 8 metabolite classes covered by our metabolomics approach, indicating a broad metabolic response to a bout of RE that expands significantly beyond energy metabolism (Figure 3 left panels).

Despite the small number of samples in our study, the changes of 7 out of these 33 metabolites were significant after correcting *p*-values for multiple testing (FDR < 0.33) (Table 1). 6 out of the 7 metabolites significantly increased in response to exercise, namely the modified amino acids 3-methylhistidine and N-lactoylvaline (Figure 3D left), the primary bile acid chenodeoxycholate, the nucleoside xanthosine (Figure 3D middle), the nicotinamide adenine dinucleotide (NAD) metabolite N1-methyl-2-pyridone-5-carboxamide and 3-(4-hydroxyphenyl)-lactate, which is a tyrosine-related metabolite considered to reduce oxidative stress in mitochondria [13]. In contrast, coenzyme-A glutathione, an antioxidant metabolite, was significantly decreased after the bout of RE (Table 1).

To identify groups of related metabolites or pathways that are induced by a bout of RE concordantly, we performed hierarchical clustering on the levels of the 33 metabolites with significant or suggestive responses (*p* < 0.05) (Figure 4). Here we highlight selected clusters of metabolites that showed such concordant patterns and might indicate specific exercise-related processes:Oxidative stress & regulation of fatty acid metabolism: CoA-glutathione, which is linked to oxidative stress response, and three medium-chain dicarboxylic fatty acids concordantly decreased post RE. Dicarboxylic fatty acids, which show antiketogenic activity, are generated through ω-oxidation of fatty acids. Their metabolism is typically upregulated in fasting state and in cases of defects of mitochondrial β-oxidation [14].Nucleotide breakdown & tissue homeostasis: Several metabolites linked to ATP deamination and NAD degradation (N1-methyl-2-pyridone-5-carboxamide (2PY)) as well as a metabolite of actin/myosin breakdown (3-methylhistidine) [15] and membrane phospholipid metabolism (glycerophosphocholine (GPC)) concordantly increased post RE. Specifically, the ATP breakdown products xanthine, hypoxanthine, as well as xanthosine were consistently increased after a bout of RE. In contrast, inosine-5′-monophosphate (IMP) was lower post resistance exercise and not part of the cluster.Protein degradation: Two amino acid derivatives (N-lactoylvaline, 3-(4-hydroxyphenyl)lactate) and 6-phosphogluconate concordantly increased after a bout of RE. N-lactoyl-amino acids are pseudo-dipeptides that are formed from lactate and amino acids by reverse proteolysis through cytosolic nonspecific dipeptidase 2, specifically when lactate or amino acid levels are increased [16]. The endogenous monosaccharide phosphate 6-phosphogluconate is a metabolite of glucose-6-phosphate, a substrate in non-enzymatic glycation processes [17].

### 2.2. Effects of Chronic Resistance Training on the Acute Response of the Muscle Metabolome

Next, we compared metabolite levels after the last bout of accustomed RE after the five-week resistance training with those after the first bout of unaccustomed RE. We found that 4 metabolites increased, whereas 42 metabolites decreased in response to chronic resistance training (*p* < 0.05) (Figure 5). These 46 metabolites are distributed over all eight analysed metabolite classes (Figure 3A,B, upper panels). Notably, 41 (4 increased; 37 decreased) of the 46 metabolites did not change in response to acute (unaccustomed) exercise. Only five metabolites changed (*p* < 0.05) both after chronic training (i.e., when comparing metabolite levels 45 min after a bout of exercise under trained versus untrained conditions) and after acute exercise (Figure 3B intersection). Levels of four of the five metabolites increased after acute exercise and showed reduced levels after long term training (e.g., xanthosine, Figure 3D middle). This can either be due to training-induced changes in the respective metabolite levels at rest between the untrained and trained muscle or be due to a training-induced reduction of the metabolites’ increase upon an exercise stimulus. Only guanosine 5′ monophosphate (5′-GMP) showed a further increase after chronic training.

The changes of 11 out of the 46 metabolites that varied in response to chronic training were significant after correcting *p*-values for multiple testing (FDR < 0.33) (Table 1). Specifically, we observed a significant increase of the glutamic acid derivative beta-citrylglutamate and significant decreases of the three N-acetylated amino acids N-acetylvaline, N-acetylleucine and N-acetylphenylalanine. Moreover, the ketone body 3-hydroxybutyrate (BHBA), the phospholipid 1-stearoylglycerophosphoethanolamine (1-stearoyl-GPE (18:0)), two metabolites of carbohydrate/energy metabolism (fructose-1,6-bisphosphate, acteylphosphate) and sulfate were all decreased. The levels of the nucleoside xanthosine and the nicotinamide adenine dinucleotide (NAD) metabolite N1-methyl-2-pyridone-5-carboxamide (2PY) increased after the first bout of exercise (acute exercise); these two metabolites were significantly lower after chronic training. This indicates either a blunted training-induced response to a bout of RE, or a lowering of metabolite levels in the resting trained muscle when compared to the resting untrained muscle.

Using the same approach as described above, we clustered the 46 metabolites with significant or suggestive effects (*p* < 0.05) in response to chronic training to highlight groups of related metabolites or pathways that are involved in concordant adaptations to resistance training:Glutamate derivatives: Beta-citrylglutamate (BCG) is a derivative of glutamic acid that was first identified in newborn rat brain and has been linked to brain development and cell-cycle regulation. It increased significantly after chronic training but did not closely cluster with any of the remaining 45 metabolites that changed in response to training (i.e., BCG showed only relatively weak concordance of changes with other metabolites), possibly indicating a unique training-related process.Energy metabolism: Fructose-1,6-bisphosphate, an intermediate of glycolysis, and acetylophosphate required for ATP synthesis concordantly decreased after chronic training.NAD metabolism: As observed in acute exercise response, sulfate and N1-methyl-2-pyridone-5-carboxamide (2PY), a metabolite of NAD turnover, change concordantly also in response to chronic training.BCAA and ketogenic amino acid metabolism: 3-Hydroxybutyrate (BHBA), a ketone body that is generated by BCAA degradation (among other pathways), and the N-acetylated BCAAs valine and leucine as well as the N-acetylated ketogenic amino acid phenylalanine concordantly decreased after chronic training.Glycerophosphoethanolamine metabolism: Four glycerophosphoethanolamines (GPEs), which are an important class of metabolites in the constellation of membrane structures, decrease concordantly in response to training. Notably, three of the four GPEs in the cluster are plasmalogens.

## 3. Discussion

Resistance exercise (RE) gained much attention to counteract sarcopenia and cachexia [18] and is considered by the WHO as an essential training mode to maintain health [19]. Yet, it is not fully understood how RE affects distinct metabolic pathways in humans. Recently, Morville and colleagues investigated differences in the time-resolved plasma metabolome responses between acute RE and EE [20]. The authors determined 833 metabolites related to lipid, carbohydrate, energy, nucleotide and amino acid metabolism and detected that more metabolites were increased after EE than in RE and, conversely, more metabolites were decreased after RE when compared to EE. Their results attribute RE as a unique exercise mode exhibiting partly a distinct metabolomic response when compared to EE. However, the plasma and blood metabolome does not always overlap, which is why the specific analysis of the skeletal muscle metabolome itself is essential [12].

Our analysis of the skeletal muscle metabolome in response to resistance exercise provides valuable novel information for understanding the relative impact of RE on muscle metabolism under different training states. Here we widened the spectrum of metabolites and metabolic pathways investigated in muscle tissue in response to acute RE, allowing us to complement previous findings in blood. For well-known RE-related phenomena such as protein degradation/damage and degradation of ATP, the effect directions of related metabolites in muscle were largely consistent with those previously observed in blood, mostly confirming these results in the relevant tissue. A notable exception was the primary bile acid chenodeoxycholate, with potential signalling function for muscle growth and atrophy [21]. We observed increased chenodeoxycholate levels in response to acute RE in muscle, while levels in blood were previously reported to be decreased [20]. We further detected changes in metabolites after repeated RE that, to date, have not been analysed in human skeletal muscle biopsies. Among these, metabolites such as beta-citrylglutamate and chenodeoxycholate may play a deeper role in the adaptive process of skeletal muscle towards exercise [21,22].

In addition to samples before and after a bout of RE in the untrained state, we compared post-exercise muscle samples of the same subjects in the unadapted and adapted states, before and after a prolonged RE intervention during which skeletal muscle had hypertrophied. Our results revealed that next to muscle growth, chronic resistance training additionally changed clusters of various intramuscular metabolites that were not directly affected by a bout of exercise in the untrained state in our study.

### 3.1. Metabolites Associated with Increased Protein Turnover after Unaccustomed RE

3-(4-hydroxyphenyl)-lactate, a metabolite of tyrosine metabolism [23], significantly increased in response to acute RE in muscle. It is known that tyrosine levels in plasma increase after exhausting exercise and elevated amino acid-levels are also associated with increased protein degradation during exercise [24,25]. Hence, the increase in 3-(4-hydroxyphenyl)-lactate may indicate augmented tyrosine metabolism in response to acute RE-induced protein degradation. Concordant with changes in 3-(4-hydroxyphenyl)-lactate, the modified amino acid N-lactoylvaline (which was misidentified as 1-carboxyethylvaline in previous versions of the applied metabolomics platform) increased in muscle tissue after acute unaccustomed RE in our study. Increased levels of this metabolite were also reported in blood plasma previously [20]. N-lactoyl-amino acids are pseudo-dipeptides in which lactate and an amino acid are connected via a peptide bond. [16] Jansen et al., recently identified this metabolite class as substrates of the ABCC5 transporter [26] and showed that their formation is catalyzed by cytosolic nonspecific dipeptidase 2 (CNDP2) in a rapid and ubiquitous process that depends on the concentrations of lactate and amino acids [16]. The abundance of N-lactoylvaline after acute exercise is in concordance with protein damage and beginning protein breakdown in muscle samples of our study [27], as both lactate and amino acid levels are high under such conditions.

In the context of protein damage, 3-methylhistidine, an established marker for protein breakdown [28], also significantly increased in muscle after RE. Most of the 3-methylhistidine is derived from skeletal muscle actin and myosin [29]. Indeed, RE induced myofibrillar damage and z-disk disruption was observed in muscle samples of our subjects [27] which is usually associated with a resulting increase in protein breakdown [30]. Proteasomal degradation as well as autophagic pathways like chaperone-assisted selective autophagy regulate the degradation and disposal of damaged skeletal muscle proteins, which we already determined in samples of this study [27,31].

In summary, changes in several metabolites detected in our study indicate increased protein breakdown in the early phase after RE.

### 3.2. Metabolites Associated with the Antioxidative System after Unaccustomed RE

We further determined that the level of CoA-glutathione (CoA-GSH) in muscle decreased in response to acute unaccustomed RE. CoA-GSH is a metabolite of the GSH antioxidative system, which—in its reduced form—is a strong antioxidant in vivo [32]. During muscular work, GSH effectively scavenges hydroxyl ions, which are increasingly generated [33], resulting in the GSH oxidation product glutathione disulfide (GSSG). GSG-reductase replenishes GSH levels by reducing GSSG under NADPH usage. This mechanism might explain why, immediately after intense RE, increased GSH levels were observed while GSSG was reduced in human blood in a previous study [34]. In rat skeletal muscle, high-intensity treadmill exercise increased GSH and GSSG whilst the ratio was maintained [35]. Analogously, CoA-GSH-reductase catalyses the reaction of the disulfide CoA-GSH to GSH and CoA [36], which may also support the maintenance of GSH, thereby reducing CoA-GSH levels as observed in our study. Based on previous findings obtained from blood plasma, our results in muscle likely reflect the impact of RE on increased oxidative stress in skeletal muscle [37].

Interestingly, CoA-GSH clustered with three medium-chain dicarboxylic fatty acid metabolites showing concordant decreases post RE. Dicarboxylic fatty acids are generated through ω-oxidation of fatty acids and their increased metabolisation is observed in the fasting state [14]; it is known that GSH preserves fatty acids from being oxidized [38]. Furthermore, the dicarboxylic fatty acid sebacate supports glucose uptake of muscle cells [39], which is important for skeletal muscle during RE and to restore substrates after exercise.

### 3.3. Metabolites That Support a Growth-Related Environment in Skeletal Muscle in Response to RE in the Trained and Untrained State

In contrast to recent observations in blood plasma [40], we saw a significant increase of the unconjugated primary bile acid chenodeoxycholate (CDCA) 45 min after RE in untrained skeletal muscle. Bile acids, mainly known for their function as detergents for ingested lipids in the intestine, have also been identified as signalling molecules, regulating energy metabolism via feedback mechanisms with endocrine factors of the fibroblast growth factor (FGF) family [41]. Regarding muscle growth after RE, FGF19 and its interrelation with CDCA is of particular interest: (i) pharmacological administration of FGF19 has been shown to induce hypertrophy in skeletal muscle of mice [42]; (ii) CDCA has been shown to induce FGF19 mRNA expression in primary human hepatocytes [43]. Considering this evidence, our finding that intramuscular CDCA levels significantly increased in response to acute RE is in line with an induction of the CDCA/FGF19 axis through RE as previously proposed. Our findings therefore oppose the assumption that a decrease of CDCA in blood plasma after RE can be transferred to muscle and indicate a downregulation of FGF19 signalling [40]. The decrease of circulating CDCA as observed by Moreville et al. might be explained by increased uptake of CDCA by muscle tissue and thus does not necessarily contradict an induction of the bile acid/FGF19 axis through RE as hypothesized by the authors. We show here that substantial differences can exist between the plasma and skeletal muscle metabolome and that the extrapolation from blood to muscle and vice versa can be misleading.

More studies are required to address the impact of CDCA and related metabolites on muscle adaptation and how plasma levels can be timely regulated compared to muscle tissue. Nonetheless, our study shows for the first time that this metabolite is likely involved in the acute regulation of the skeletal muscle environment under conditions of tissue damage and muscle growth.

Beta-citrylglutamate (BCG) significantly increased after chronic training, i.e., when comparing metabolites after a bout of exercise between the unadapted and the adapted state. Though known for a while, the actual physiological role of beta-citrylglutamate is still elusive. It has been shown that BCG is transported in vivo by the ABCC5 transporter, which affects the disposition of endogenous metabolites, toxins and drugs [26]. It was first identified in newborn rat brain and testes and has been linked to brain development [44]. The brain level of BCG is the highest when the proliferation rate of neurons in cerebral cortex is high and it decreases when neurons mature [45]. In primary cultures of neurons from newborn mouse brain, BCG, which was suggested to serve as an Fe-carrier for aconitase [44], enhanced cell viability by accelerating mitochondrial activity. Rats that are infertile due to germ cell depletion show low beta-citrylglutamate concentrations, suggesting its involvement in the metabolic support of cell proliferation [46]. BCG is a physiological substrate of the enzyme beta-citrylglutamate synthase-B, encoded by the RIMKLB gene [47]. In humans, it is mainly expressed in the testes but also in other tissues, including skeletal muscle [48]. Genetic variants in this gene were found to be associated with BCG (X-12748) levels in blood [49]. Interestingly, a meta-analysis of human muscle biopsy studies showed that RIMKLB expression increased by 43% (log2 0.52) after a bout of RE [50]. Therefore, it may be hypothesized that increased levels of BCG could support satellite cell responses that accompany skeletal muscle growth conditions in shorter time frames than applied in our intervention [51]. While we did not observe an increase of BCG (*p* < 0.05) after acute exercise in our study, a closer inspection of the individual metabolite profiles revealed that, in fact, four of six participants did show an increase in BCG (Appendix A). 

From the first unaccustomed bout to the last bout after chronic RE training, BCG levels increased in all subjects, including those for whom no increases were detected in acute unaccustomed exercise. This could indicate that repeated stimulation of muscle increasingly involves the synthetic pathway of BCG. However, as no second baseline biopsy was taken after repeated training, we cannot definitively say whether resting levels were also higher after 5 weeks or the acute response after RE was augmented.

### 3.4. Metabolites Reflecting Acutely Increased Energy and Nucleotide Metabolism in Response to RE in the Trained and Untrained State

In metabolically active tissues, levels of NAD^+^/NADH and nucleotides are tightly linked to each other and ensure high rates of ATP turnover [52]. Hence, it is comprehensible that we observed changes in metabolites related to nucleotide and NADH turnover also in skeletal muscle. Increased ATP breakdown due to EE but also RE exerts a rise in adenine nucleotide metabolites [53]. Those metabolites are metabolized within the purine nucleotide cycle to hypoxanthine, xanthine and xanthosine [54], which were shown to be elevated up to 180 min after EE and RE in plasma [20]. Hence, our study reveals that muscle is the origin for changes in blood of those nucleotides. Generally, the concentrations of nucleotide degradation products increase most after intense muscle contractions. Acute RE increases xanthine oxidase levels from 45 min up to 96 h in human skeletal muscle [55]. Xanthine oxidase converts hypoxanthine to xanthine. Therefore, the strong increases in these metabolites, including xanthosine, likely reflect the impact of acute RE on adenine-nucleotide metabolism.

Xanthine can be further metabolized to IMP and xanthosinemonophosphate (XMP) [56]. Interestingly, IMP decreased after acute RE in the unaccustomed state (Figure 4). Assuming that IMP is rapidly metabolized within 45 min after RE in muscle, this may explain why IMP levels declined while xanthine was still elevated. Indeed, lower IMP levels were also observed in human skeletal muscle acutely after sprint training [57].

Xanthosine was significantly increased after the first bout of acute RE. Comparing the levels after a single bout of RE in the trained versus the untrained state, xanthosine was lower after the 5-week training, which could indicate a reduced increase of levels in response to a bout of RE after training or a reduced level at rest after training.

The nicotinamide adenine dinucleotide (NAD) metabolite N1-methyl-2-pyridone-5-carboxamide (2PY) is a metabolite of NAD-degradation [58]. Catabolism of NAD increases significantly during exercise when oxidative metabolism increases [59]. NAD is involved in ATP production and its turnover can be measured by the urinary outputs of 2PY and 4PY. Increased levels therefore likely reflect an acutely increased NAD turnover induced by an unaccustomed bout of RE. Increased levels of 2PY were detected in urine of mice subjected to a high fat diet and an associated increase of oxidative metabolism suggesting that 2PY reflects oxidative capacity [60]. Endurance exercise significantly enhances oxidative capacity of mitochondria and the rate of NAD turnover [59]. So far, specific changes in 2PY metabolite levels in human skeletal muscle in response to RE have not been investigated. However, it has been shown that prolonged RE increases NAD and NADH levels in human subjects [61]. Similar to xanthosine, we observed a decrease of 2PY when comparing the post-exercise levels in the trained versus the untrained state. This response may be explained by the reduced activity of enzymes involved in the generation of 2PY, a generally reduced requirement for NAD turnover after training or also an increased speed of 2PY excretion, which may decline muscle 2PY levels.

### 3.5. Metabolites Reflecting Changes in the Skeletal Muscle Lipid Profile after a Period of Resistance Training

Increased fatty acid metabolism has been shown to occur during and after EE but also after RE in skeletal muscle [62,63]. However, it is not clear in which direction repeated RE adapts lipid-derived metabolites in skeletal muscle. We detected that a cluster of phospholipids, mostly consisting of metabolites of glycerophosphoethanolamine plasmalogens, were reduced after RE in the trained state (Figure 5). Plasmalogens are a unique class of glycerophospholipids and important components of membrane structures and are enriched in the kidney, lung and skeletal muscle [64,65]. Although increased metabolism of plasmalogens in resistance-exercising human skeletal muscle to date have not been specifically determined, RE-induced changes could be assumed in a tissue that is constantly stimulated for tissue growth and challenged to maintain proteostasis.

### 3.6. Metabolites Reflecting the Modulation of Skeletal Muscle Energy Metabolism after a Period of Resistance Training

Chronic RE training reduced fructose 1-6-bisphosphate (FBP) and acetylphosphate levels compared to unaccustomed exercise in our study. While FBP is a glycolytic intermediate generated by phosphofructokinase (PFK), acetylophosphate resynthesizes ATP. Chronic RE can increase the oxidative capacity of skeletal muscle while its glycolytic capacity decreases [66]. A reduction in glycolytic capacity in combination with increased oxidative capacity may have reduced levels of FBP in our study by a reduction in PFK activity and increased oxidative metabolisation of glycolytic products. We can only speculate about those events since we have not measured the levels or activity of the related enzymes.

### 3.7. Metabolites Reflecting Changes in the Profile of N-Acetylated Ketogenic Amino Acids after a Period of Resistance Training

After chronic training, we observed decreased levels of N-acetylated ketogenic amino acids (leucine, phenylalanine) and valine and the ketone body 3-hydroxybutyrate (BHBA). Usually, RE significantly augments the uptake of leucine, which then stimulates mTORC-1 activity and, consequently, the early RE-induced increase in myofibrillar and sarcoplasmic protein synthesis [67]. Because expression of amino acid transporters is augmented in response to acute RE [68], increases in metabolites associated with BCAA metabolism would have been expected in skeletal muscle. Instead, we detected a reduced abundance of their N-acetylated forms. Acetylation is a very common physiological mechanism which alters the function of proteins. Acetylated proteins are known to increase in skeletal muscle in response to exercise [69]. Intense exercise has been shown to change the acetylation of histones and mitochondrial proteins [70,71]. Importantly, N-acetylated amino acids e.g., N-acetylleucine have opposite roles than the non-acetylated forms, and N-acetylleucine has been shown to block p70s6k activation and induce cell cycle arrest in cells [72]. In this regard, it may be beneficial for muscle adaptation when levels of N-acetylated amino acids are reduced. However, it is still unclear what their fate in skeletal muscle is and whether those amino acids are increasingly metabolized or the general acetylation is blunted. Another pathway that is activated by RE and involved in the regulation of protein synthesis and muscle hypertrophy is the cancer-like reprogramming inducing the synthesis of serine and glycine via PHGDH [8,73]. RE stimulates the expression of pyruvate kinase muscle 2 (PKM2), which in C2C12 myotubes has been shown to induce hypertrophy [74]. The present results do not show a significant increase in serine and glycine levels; however, this could be due to the fact that these metabolites are immediately incorporated in the synthesized contractile and cytoskeletal proteins.

Interestingly, BHBA (3-hydroxybutyrate) significantly decreased after chronic training concordantly with the N-acetylated amino acids in our study. BHBA is a ketone body originating from lipid or protein metabolism and is increased during starvation or endurance exercise [75]. It serves as a fuel source in skeletal muscle, and some authors recognized a correlation between BHBA levels as well as skeletal muscle function and cognitive capacity [76]. Interestingly, endurance trained humans have a significant greater capacity to oxidize ketones during exercise and therefore show reduced levels after exercise [77]. Based on this observation, reduced levels in our study may reflect either a training induced increase in the capacity to oxidize BHBA at the end of the study or also a chronic reduction in baseline levels.

### 3.8. Limitations

While we see the use of human samples as a particular strength of our study, we must acknowledge that our sample size is small, limiting statistical power. Furthermore, 45 min post RE was the only time point where we collected muscle biopsies after acute resistance exercise. At this time point, protein synthesis and degradation as well as restoration of tissue homeostasis and integrity is still being regulated [27,30]. Indeed, intense exercise stimuli affect molecular events for several days after stimulation. It can also be assumed that several hours after stimulation, the metabolome is still changing. Therefore, we have detected only a snapshot of a changed intramuscular metabolome and further metabolites that may contribute to increased muscle anabolism and hypertrophy might not have been detected at this time point [78]. A further limitation was that we did not collect a second baseline biopsy at the end of the training period. As a consequence, we are not able to clearly differentiate whether changes in response to chronic training occur (i) due to changes in resting levels between the untrained and trained state (while the acute exercise-induced change remains the same between states) or (ii) due to changes in the steepness of increase or decrease in response to the acute exercise stimulus between the untrained and trained states (while the resting levels remains the same between states). Finally, whilst the measurement of steady state metabolite concentrations is useful to identify changes in the network of metabolic pathways, it does not inform about actual metabolic fluxes. 

## 4. Methods

### 4.1. Ethics Statement

Participants were informed verbally and in writing of the study’s purpose and the possible risks involved before providing written informed consent to participate. The study was approved by the Ethics Committee of the German Sport University Cologne in compliance with the Declaration of Helsinki (Approval code: 10/2013).

### 4.2. Subjects

14 moderately resistance-exercise-trained male subjects took part in a resistance-exercise study designed to investigate other scientific aspects of skeletal muscle biology [27,31]. From six subjects, sufficient muscle samples were left for metabolomic analysis in this approach (age: 24 ± 4 years; height: 183 ± 9 cm; weight: 81 ± 11 kg) (Appendix A). All subjects did not perform resistance exercise for more than two times per week prior the intervention. All participants were recruited from the German Sports University Cologne via the Internet and local information in study courses.

### 4.3. Study Design

Subjects were instructed to refrain from resistance exercise 14 days prior to the intervention and from any physical activity 48 h prior to baseline biopsies, exercise testing and each resistance exercise session. Subjects participated in a three times weekly (Monday, Wednesday and Friday) resistance exercise regimen containing 13 subsequent resistance exercise sessions in five weeks. Skeletal muscle biopsies were collected 72 h before the first training at rest (rest), 45 min after the first bout of exercise (first bout of RE) and 45 min after the 13th exercise session (last bout of RE) to analyse changes in the muscle metabolome and myofibre diameter via immunohistochemistry. 

### 4.4. Resistance Exercise Sessions

The study protocol is illustrated in Figure 1. Prior to the study, strength tests were conducted to adjust the resistance exercise load to match the 10-repetition maximum (10RM) desired in the training sets. On the day of the training sessions and the biopsies, subjects reported to the laboratory between 07:30 and 08:30 AM. Prior to the resistance exercise bouts, a five-minute warm-up with a workload of 1 W/kg on a cycle ergometer was performed (see “Exercise testing”). Following that, subjects were allowed to conduct one single set of resistance exercises with eight repetitions and approximately 50% of 10RM to conduct the resistance-exercise-specific warmup. Training started and was conducted finally between 08:00 AM and 09:30 AM. Each training session lasted for 45 min, including warmup. During resistance exercise sessions, participants conducted six sets of leg resistance exercises (three sets of leg extension followed by three sets of leg press with two min rest between each set and three min rest between exercises) using a standard leg extension and leg press machine (Gym 80, Gelsenkirchen, Germany). Subjects aimed to perform resistance exercises with 10 repetitions per set. Due to fatigue from set to set and adaptation from session to session, subjects trained in a range between 8 and 12 repetitions. Each repetition followed a standardized contraction pattern consisting of a sequence of two seconds concentric, one second isometric and two seconds eccentric contractions. Training intensity (weight loading) was increased in the following resistance exercise session when subjects were able to perform more than 12 repetitions in one exercise set. This was conducted to ensure adaptation on skeletal muscle over time. Each resistance exercise session was supervised and documented by scientific co-workers of the German Sports University Cologne participating in the project. At a time of 60 min before each resistance exercise session and 120 min before the resting biopsy, subjects were advised to consume a standardized protein energy drink (Fresubin^®^ protein energy drink, Fresenius Kabi Deutschland GmbH, Bad Homburg, Germany; containing 20 g protein, 24.8 g carbohydrate, 13.4 g fat, providing 1260 kJ) As the training regimen lasted around 45 min, post exercise biopsies were collected 150 min after drink ingestion. The time frame was chosen to ensure that no subject was in the fasted state but also to reduce acute feeding-induced changes of muscle metabolism [79]. The drink was provided to the subjects beforehand and they were advised to stay fasted overnight from 22:00 until 60 min before resistance exercise training the next morning.

### 4.5. Skeletal Muscle Biopsies

Resting skeletal muscle biopsies (rest) from musculus vastus lateralis were taken between 5 and 7 days before the intervention. Post-exercise biopsies were collected 45 min after the first bout of unaccustomed resistance exercise (first bout of RE) and 45 min after the last (13th) resistance exercise-session (last bout of RE). Fresh skeletal muscle tissue was freed from blood and visible fat or connective tissue was removed. Muscle tissue was immediately placed in cryotubes and frozen in liquid nitrogen. Tissue samples for immunohistochemistry were carefully aligned for cross-sectional analysis, embedded in tissue freezing medium, frozen in pre-cooled isopentane and stored at −80 °C for further analysis.

### 4.6. Immunohistochemistry and Determination of Myofibre Diameter

Consecutive 7 µm cross-sections of each biopsy time point were cut with a LEICA CM7300 Cryostat and mounted on Polysine^®^ slides (VWR International GmbH, Darmstadt, Germany) air-dried and stored at −80 °C until further analysis. Slides were initially brought to room temperature and afterwards incubated for eight min in −20 °C acetone. Sections were then blocked in 0.05 mM TBS containing 5% bovine serum albumin (BSA) for one hour at room temperature. To detect type I and type II fibres and determine the borders of myofibres at the basement membrane, cross sections were incubated overnight at 4 °C with monoclonal mouse primary antibodies (Developmental Studies Hybridoma Database (Iowa City, IA, USA) and raised against human adult slow myosin heavy chain type I (A4.951) and dystrophin (MANEX1011B). Antibodies were diluted 1:200 and 1:25 in 0.8% BSA-TBS, respectively. On the following morning, slides were rinsed five times for five min with TBS and then incubated for one hour with goat-anti mouse polyclonal biotinylated secondary antibodies (Dako Cytomation, Glostrup, Denmark) diluted 1:400 in 0.05 mM TBS. Slides were then incubated for one hour with Streptavidin biotinylated Horseradish Peroxidase complex (Amersham Biosciences, Uppsala, Sweden) diluted 1:400 in TBS. Staining was carried out using a 3,3’-diaminobenzidine (DAB) solution (0.09 M phosphate buffer (pH 7.4), 2.2 mM DAB, 7.03 mM ammonium chloride, 0.93 mM nickel sulfate, 10.44 mM ß-D-glucose and 0.000024 mM glucose oxidase). Hereafter, the staining procedure was repeated. Cross-sections were then incubated overnight with primary antibodies raised against type I and IIA fibres (N2.261; Developmental Studies Hybridoma Database (Iowa City, IA, USA)) diluted 1:100 in 0.8% BSA-TBS. All other procedures were identically repeated, but staining was then carried out using a HRP-based solution (HRP-green solution; 42 Life Sciences, Bremerhaven, Germany), staining type IIA fibres in green but leaving IIX fibres unstained. After a short dehydration of the cross sections (30 s incubation with 80% Ethanol, 30 s incubation with 100% Ethanol and 30 s incubation with Xylol), stained cross-sections were embedded in Entellan (Merck, Darmstadt, Germany) and covered with a coverslip. Muscle cross-sections from all biopsy time points of subjects and groups were stained within a single batch using the same antibody dilution and development time to minimize variability in staining efficiency. Five to seven digital photos of each cross section were captured in 20-fold magnification via a Zeiss KS-300 light microscope equipped with a digital CCD Camera (Sony, Tokyo, Japan). By applying the specific pixel/aspect ratio of the used 20× objectives (2.4 pixel per µm), the best fitting ellipse tool using the software ImageJ^®^ (National Institutes of Health, Bethesda, MD, USA) was applied to determine the inner borders of selected myofibres as the minor axis. Thirty-five myofibres per fibre type (type I and type II), time point and subject were analysed for the minor axis. As type IIX fibres could not be found in every subject, IIX fibres were excluded from myofibre diameter analysis.

### 4.7. Tissue Homogenization

Frozen muscle samples in a weight range of 17 to 58 mg were weighted and placed in pre-cooled (dry ice) 2 mL homogenization tubes containing ceramic beads with a diameter of 1.4 mm. Pre-cooled water with a ratio of 15 µL/mg tissue was added into the tubes. The samples were then homogenized in a Precellys 24 homogenizer (PEQLAB Biotechnology GmbH, Erlangen, Germany) equipped with an integrated cooling unit 6 times for 20 s at 5500 rpm, with 30 s intervals (to ensure freezing temperatures in sample vials) between the homogenization steps. After homogenization, 100 µL of the homogenate was loaded onto a 2 mL 96-deep well plate.

### 4.8. Metabolomics Measurement

The 100 µL homogenate in the 2 mL 96-deep well plate was extracted with 475 µL methanol containing four recovery standards to monitor the extraction efficiency. After centrifugation, the supernatant was split into 4 aliquots in two 96-well microplates. The first two aliquots were used for ultra-high performance liquid chromatography-tandem mass spectrometry (UPLC-MS/MS) analysis in positive and negative electrospray ionization mode. The samples were dried on a TurboVap 96 (Zymark, Hartwell, GA, USA). Prior to LC-MS/MS measurements, the samples were reconstituted with 80 µL of 0.1% formic acid for the positive ion mode, and with 80 µL of 6.5 mM ammonium bicarbonate pH 8.0 for the negative ion mode. UPLC-MS//MS analysis was performed on a Q Exactive high resolution/accurate mass spectrometer (Thermo Fisher Scientific GmbH, Dreieich, Germany) coupled with an Acquity UPLC system (Waters GmbH, Eschborn, Germany). Two separate columns (2.1 × 100 mm Waters BEH C18 1.7 µm particle) were used for acidic (solvent A: 0.1% formic acid in water, solvent B: 0.1% formic acid in methanol) and for basic (A: 6.5 mM ammonium bicarbonate pH 8.0, B: 6.5 mM ammonium bicarbonate in 95% methanol) mobile phase conditions, optimized for positive and negative electrospray ionization (ESI), respectively. After injection of the sample extracts, the columns were developed in a gradient of 99.5% A to 98% B In 11 min run time at 350 µL/min flow rate. The eluent flow was directly connected to the ESI source of the Q Exactive mass spectrometer. Full scan mass spectra (80–1000 *m*/*z*) and data dependent MS/MS scans were recorded in turns. Metabolites were annotated by curation of the LC-MS/MS data against Metabolon’s proprietary chemical database library (Metabolon, Inc., Durham, NC, USA) based on retention index, precursor mass and MS/MS spectra. The metabolites were assigned to cellular pathways based on PubChem, KEGG, and the Human Metabolome Database. In total, 645 metabolites from different metabolite classes (amino acids (and derivatives), peptides, nucleotides (and derivatives), carbohydrates, energy metabolites, lipids, xenobiotics, cofactors and vitamins) were measured. Thereof, 508 metabolites were annotated with a chemical name, while 137 metabolites represented molecules with unidentified chemical structure (unknown metabolites). The full list of metabolites is provided in Appendix A.

### 4.9. Statistical Analysis

Raw ion counts of the 617 metabolites that showed less than 70% missing values across all samples were log2 transformed to achieve approximation of metabolite abundance distributions to normal distribution. Paired *t*-tests were applied to each metabolite and two pairs of time points based on the log2 transformed metabolite dataset. First, we compared metabolite levels at rest versus those post one (first) bout of exercise to assess the effects of acute exercise. Second, we compared the levels post first bout versus post last bout of exercise after 5 weeks of resistance training to assess the effects of chronic training. To adjust for multiple testing of 617 metabolites, we controlled the false discovery rate at a level of 1/3 (FDR < 0.33; i.e., accepting up to 1/3 expected false positive results among the reported hits) using the calculation of *q*-values [80]. Using this approach, 7 and 11 metabolites were significantly changed in response to acute exercise and chronic training, respectively (with accepting 2 out of the 7 and 4 out of the 11 metabolites to be potentially false positives). Metabolites with a raw (uncorrected) *p*-value < 0.05 were considered as suggestive results. Log2 fold changes as well as *p*- and *q*-values for the full list of metabolites are provided in Appendix A (acute exercise) and Appendix A (chronic training). Additionally, we summarized results in volcano plots, which relate the log2 fold change of metabolites between time points to the -log10 *p*-value resulting from the *t*-test for all metabolites. Metabolites showing suggestive effects (*p* < 0.05) were selected for visualization and clustering using heatmaps (Bioconductor ComplexHeatmap R package [81]. To this end, z-scores of metabolite levels (i.e., scaling values to a mean of 0 and a standard deviation of 1) were calculated based on the complete data set, including all participants and time points per metabolite. In the heatmaps, these z-scores are visualized for each participant and time point using a color gradient going from blue (low levels) via white (average levels) to red (high levels). Metabolites are grouped according to hierarchical complete linkage clustering on Euclidean distances of metabolite z-scores across samples.

Statistical analyses for metabolites and plots were performed using R 3.5.1 (R Foundation for Statistical Computing; version 3.5.1). To determine differences in fibre type diameter for type I and II myofibers, we performed a paired *t*-test. Significance was accepted with an alpha-level below 5% (*p* < 0.05).

## 5. Conclusions

From our study, we conclude that the skeletal muscle metabolome is sensitive towards acute RE in the trained and untrained state and reflects the adaptive process of skeletal muscle towards repeated stimulation. Our findings shed more light on the broad response of skeletal muscle metabolism after a bout of unaccustomed resistance exercise and after an additional period of resistance training, ranging from energy metabolism to tissue remodeling. Related metabolites such as beta-citrylglutamate, which can also be detected in plasma, could serve as new biomarkers to estimate skeletal muscle adaptation, supporting an easy and non-invasive future diagnostic procedure during extended training phases in athletes and patients.

## Figures and Tables

**Figure 1 metabolites-12-00445-f001:**
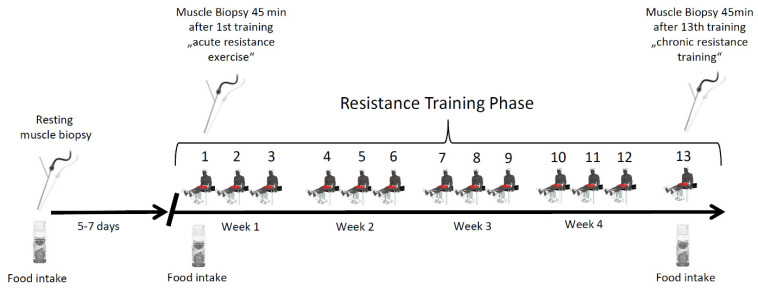
Study protocol. Schematic depiction of the study protocol. Subjects conducted 13 resistance exercise sessions over the time course of the study. Muscle biopsies were taken at rest as well as 45 min after the first and last training session in the fed state.

**Figure 2 metabolites-12-00445-f002:**
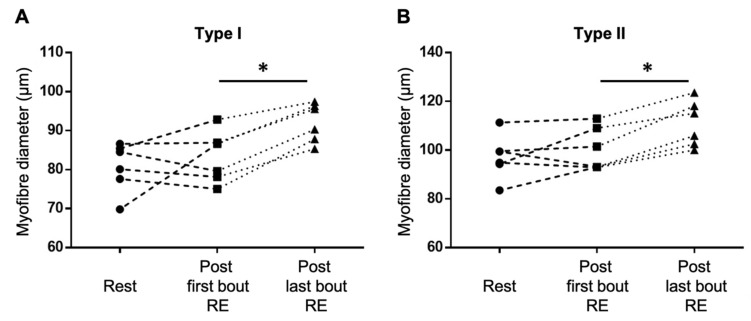
Five weeks of resistance training induced a skeletal muscle fibre hypertrophy of both type I (**A**) and type II (**B**) muscle fibres. * *p* < 0.05.

**Figure 3 metabolites-12-00445-f003:**
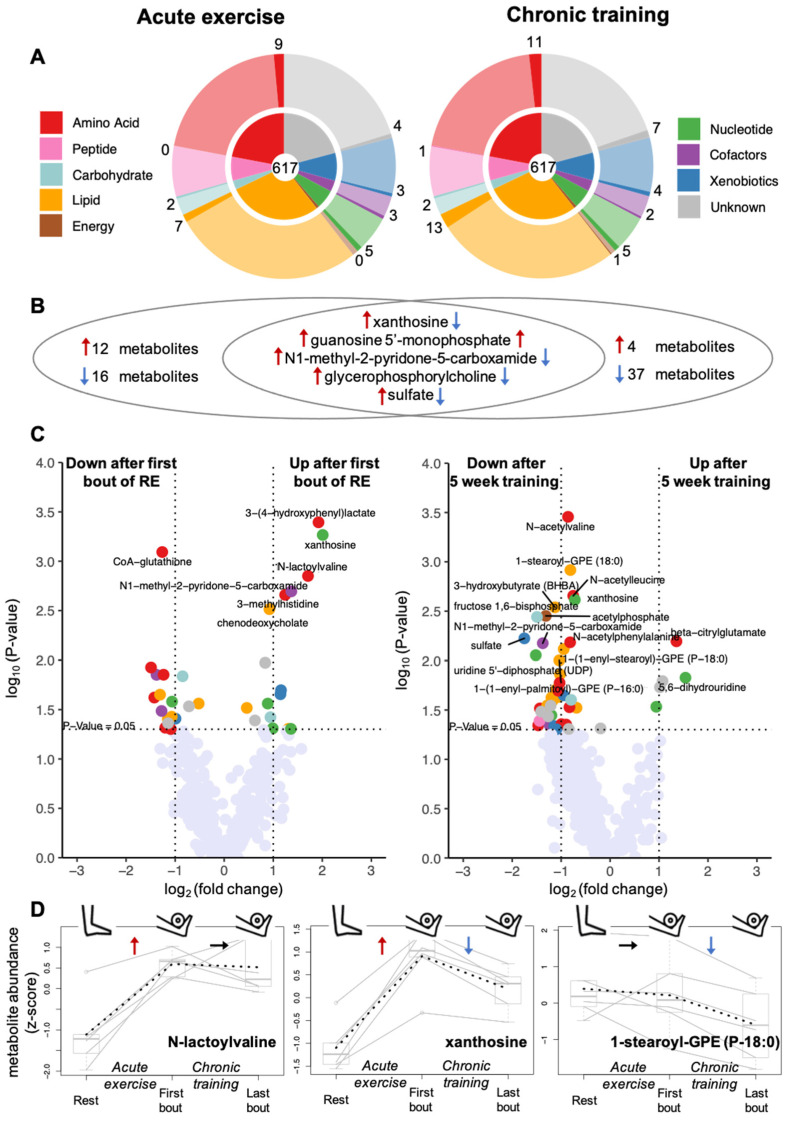
Effects of acute exercise and chronic training on muscle metabolome. (**A**) The 617 analysed metabolites belong to nine different metabolite classes; the inner circles of the pie charts show the fraction of metabolites falling into each class; the fraction of affected metabolites (*p* < 0.05) per class is marked by saturated colour in the outer rings. (**B**) In total, acute exercise (i.e., comparing metabolite levels after first bout of exercise with levels at rest) and chronic training (i.e., comparing levels after last bout of exercise after 5 weeks training with levels after first bout of exercise) showed an effect on 33 (left) and 46 (right) metabolites, respectively, with 5 metabolites being changed in response to both conditions (middle); red arrows indicate an increase, blue arrows a decrease in metabolite levels. (**C**) Volcano plots displaying the *p*-values (−log10) versus log2 fold changes of all metabolites after acute exercise (left) and after chronic training (right); metabolites with *p* < 0.05 are shown in the color of their metabolite class using the same color coding as in A. (**D**) Metabolite levels for all 6 participants at the three sampling points shown for three selected metabolites with typical patterns of changes after acute exercise (N-lactoylvaline; left), chronic training (1-stearoyl-GPE (P-18:0); right), and both conditions (xanthosine; middle), respectively; solid lines connect the levels of each individual; dotted lines connect the mean levels at each sampling time point.

**Figure 4 metabolites-12-00445-f004:**
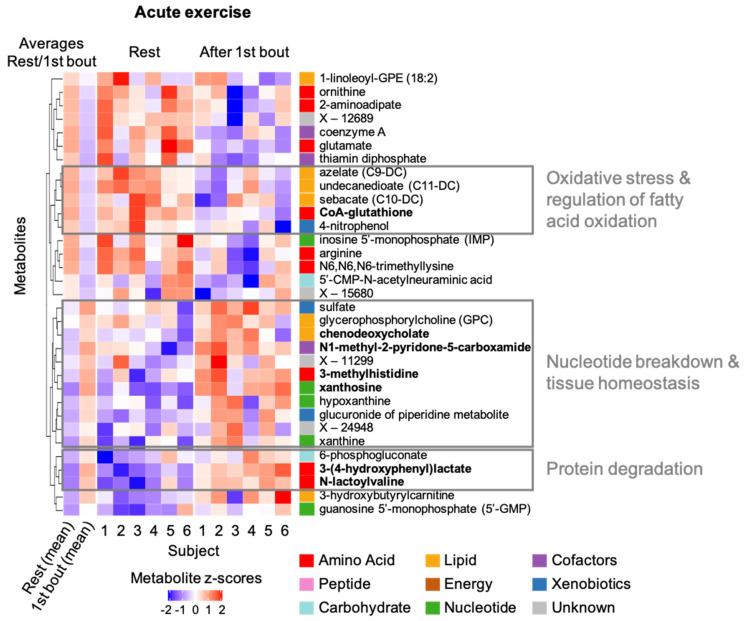
Heatmap of metabolites changing in response to acute exercise. Individual and average metabolite levels at rest and after the first bout of exercise are shown as z-scores for all 33 metabolites with changes in response to acute exercise (*p* < 0.05); red color indicates higher levels compared to all levels measured for the specific metabolite in the present study; analogously, blue color indicates lower levels; names of metabolites with significant changes after correcting for multiple testing are shown in bold. Metabolites were clustered by their similarity in metabolite levels at the two conditions across the participants; the resulting clusters are displayed in the dendrogram on the left of the heatmap. Selected clusters and their link to exercise-related molecular processes are highlighted.

**Figure 5 metabolites-12-00445-f005:**
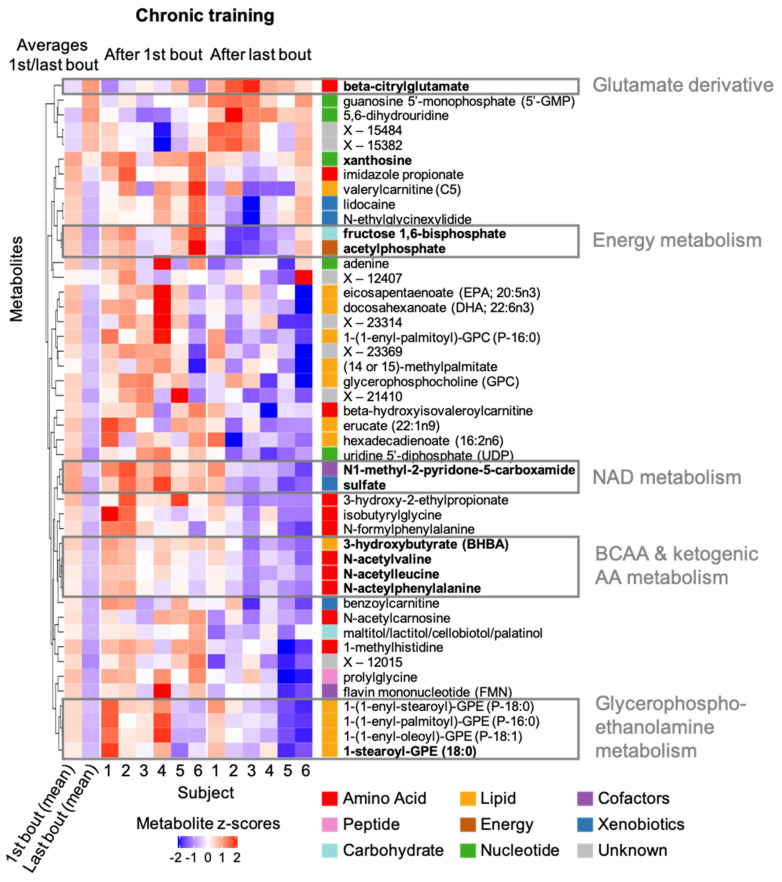
Heatmap of metabolites changing in response to chronic training. Individual and average metabolite levels after the first bout of exercise and after the last bout of exercise following five-week resistance training are shown as z-scores for all 46 metabolites with changes (*p* < 0.05) in response to chronic training; red color indicates higher levels compared to all levels measured for the specific metabolite in the present study; analogously, blue color indicates lower levels; names of metabolites with significant changes after correcting for multiple testing are shown in bold. Metabolites were clustered by their similarity in metabolite levels at the two conditions across the participants; the resulting clusters are displayed in the dendrogram on the left of the heatmap. Selected clusters and their link to exercise-related molecular processes are highlighted. BCAA: branched chain amino acid; AA: amino acid; NAD: nicotinamide adenine dinucleotide.

**Table 1 metabolites-12-00445-t001:** Metabolites with significant changes after acute exercise and/or chronic training. Bold font indicates significance controlled by false discovery rate; provided *p*-values are uncorrected.

Metabolite	Metabolite Class	Acute Exercise Log2 Fold Change	Acute Exercise *p*-Value	Chronic Training Log2 Fold Change	Chronic Training *p*-Value
3-(4-hydroxyphenyl)-lactate	Amino Acid	1.92	4.03 × 10^−4^	−0.43	0.38
CoA-glutathione	Amino Acid	−1.26	8.07 × 10^−4^	−0.02	0.96
N-lactoylvaline *	Amino Acid	1.71	1.41 × 10^−3^	−0.08	0.82
3-methylhistidine	Amino Acid	1.24	2.19 × 10^−3^	−1.18	0.12
N-acetylvaline	Amino acid	0.12	0.87	−0.86	3.50 × 10^−4^
N-acetylleucine	Amino acid	−0.16	0.83	−0.75	2.21 × 10^−3^
beta-citrylglutamate	Amino acid	0.41	0.32	1.36	6.39 × 10^−3^
N-acetylphenylalanine	Amino acid	−0.35	0.62	−0.82	6.54 × 10^−3^
chenodeoxycholate	Lipid	0.92	3.06 × 10^−3^	−0.30	0.62
1-stearoyl-GPE (18:0)	Lipid	−0.18	0.63	−0.81	1.21 × 10^−3^
3-hydroxybutyrate (BHBA)	Lipid	0.28	0.71	−1.13	2.89 × 10^−3^
xanthosine	Nucleotide	2.01	5.41 × 10^−4^	−0.72	2.43 × 10^−3^
N1-methyl-2-pyridone-5-carboxamide	Cofactor & vitamin	1.37	2.02 × 10^−3^	−1.38	6.68 × 10^−3^
acetylphosphate	Energy	0.37	0.25	−1.31	3.53 × 10^−3^
fructose-1,6-bisphosphate	Carbohydrate	0.54	0.14	−1.49	3.62 × 10^−3^
sulfate	Xenobiotics	1.16	0.020	−1.75	5.97 × 10^−3^

* misidentified as 1-carboxyethylvaline in previous versions of the applied metabolomics platform.

## Data Availability

The data that support the findings of this study are available from the corresponding author upon reasonable request. The data that supports the findings of this study are available in the Appendix A of this article.

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
