# Peer review of "Effects of Acute and Chronic Resistance Exercise on the Skeletal Muscle Metabolome"

_metabolites, 2022, doi:10.3390/metabo12050445_

Round 1
Reviewer 1 Report
Very interesting manuscript. It is one more step for revealing the signaling mechanisms for resistive exercise adaptive effects. The authors tested the acute metabolome markers after one bout of resistive exercise in human. Thery also investigated the acute effects of the exercise bout after long term training period. The results are interesting but as for the long-term effects they can not be called the chronic effects, since the samples were harvested only 45 min after the 13th exercise bout. This note should be considered for the title of the article and for the main body of the text. I guess that the authors understand this problem and they describe it in the "Limitation" paragraph but they should exclude all the word regarding "chronic" effects. They woul get chronic effects if they take biopsy after 1-3 days after the last bout (better 2 days).
This is my main concern.
Author Response
Dear Madame or Sir,
we highly appreciate your review and especially your comment on the chronic effects. We work a lot on human muscle biopsy samples after acute resistance exercise but also chronic resistance training. When you investigate the muscle after an acute stimulation by resistance exercise there is an extended regulation of many processes and also protein synthesis up to 72 hours after acute RE. Therefore, it is difficult to be sure that there is a recurring homeostasis within 2-3 days. However, considering your proposal, we would then have likely detected a more stable return to baseline situations after a chronic training period, that is correct. However, we aimed to analyze the acute response of the metabolome after an extended or “chronic” RE training, a stimulation pattern which induces structural adaptations e.g. hypertrophy. This is what can be expected only after chronic, but not acute (e.g. a single) training.
When writing this manuscript, we thought several times about the best wording for our approach but finally considered to stick to “chronic”, as it also improves the readability compared to “repeated RE training” etc. vs “acute RE training” in this manuscript. This, because there are basically two acute responses.
We made sure that this will be especially clear, as we introduced the specificities of our approach several times in the introduction, the methods and the discussion of the manuscript. We also explained our exercise approach by adding several times the wording “accustomed” and “unaccustomed”, to make clear, that there is a training phase where muscle also structurally adapts to chronically repeated RE training.
Therefore, we have here the situation where chronic training adapts skeletal muscle and we concomitantly analyzed the acute changes of the metabolome under these conditions.
We really would appreciate to keep to this wording but also changed at further spots (line 97 ,101, 145 and line 199) the text and the headlines to make it more clear what we did.
Sincerely yours,
Sebastian
Reviewer 2 Report
Reviewer Report
This study is very important and adds knowledge on muscle metabolism. The authors provide novelties regarding the impact of resistance exercise on human skeletal muscle metabolism. They approached a mass spectrometry-based metabolomics evaluating biopsies from vastus lateralis muscle. The manuscript is very well written and very clear. Results are very interesting and well discussed. I consider useful the discussion on the different results obtained for some metabolites in muscular tissue rather than in blood samples.
However, I suggest
- Editing improvement is required throughout the text.
- The number of the ethic committee (line 522) should be mentioned.
Author Response
Dear Madame or Sir,
We thank you for your response and the time for reviewing our paper.
We have edited the manuscript thoroughly and also edited the number of the ethics approval code in the methods.
Yours sincerely,
The authors
Round 2
Reviewer 1 Report
The authors explained their considerations regarding the usage of the words "chronic effects" in the manuscript. They changed some statements.